# The DeepMotion entry to the GENEA Challenge 2022

Shuhong Lu
Andrew Feng
shuhongl@usc.edu
feng@ict.usc.edu
Institute for Creative Technologies, University of Southern California
Los Angeles, California

## ABSTRACT

This paper describes the method and evaluation results of our Deep-Motion entry to the GENEA Challenge 2022. One difficulty in data-driven gesture synthesis is that there may be multiple viable gesture motions for the same speech utterance. Therefore the deterministic regression methods can not resolve the conflicting samples and may produce more damped motions. We proposed a two-stage model to address this uncertainty issue in gesture synthesis. Inspired by recent text-to-image synthesis methods, our gesture synthesis system utilizes a VQ-VAE model to first extract smaller gesture units as codebook vectors from training data. An autoregressive model based on GPT-2 transformer is then applied to model the probability distribution on the discrete latent space of VQ-VAE. The user evaluation results show the proposed method is able to produce gesture motions with reasonable human-likeness and gesture appropriateness.

## CCS CONCEPTS

• **Computing methodologies** → **Intelligent agents**; **Animation**; *Neural networks*.

## KEYWORDS

gesture synthesis, computer animation, neural networks

**ACM Reference Format:**
Shuhong Lu and Andrew Feng. 2022. The DeepMotion entry to the GENEA Challenge 2022. In *INTERNATIONAL CONFERENCE ON MULTIMODAL INTERACTION (ICMI '22), November 7–11, 2022, Bengaluru, India.* ACM, New York, NY, USA, 7 pages. https://doi.org/10.1145/3536221.3558059

## 1 INTRODUCTION

Producing realistic non-verbal behaviors that mimic human behaviors is vital for a virtual human to communicate effectively in a social interaction with human users. Co-speech gestures synthesis from speech audio, therefore, plays an important role in creating an effective embodied agent since it is not feasible to manually create gesture motions for all speech utterances. Such capability will find applications in areas such as games, education, and virtual

*ICMI '22, November 7–11, 2022, Bengaluru, India*
© 2022 Association for Computing Machinery.
ACM ISBN 978-1-4503-9390-4/22/11...$15.00
https://doi.org/10.1145/3536221.3558059

reality. However, synthesizing realistic 3D gesture motions from only speech audio is still a challenging and unsolved problem.

One challenge in learning a model for gesture synthesis is that there may be multiple viable gesture motions for the same speech utterance. Therefore a direct regression method may not be able to learn the correct speech-to-gesture mapping when there are conflicting examples in the dataset. Since such methods formulate the prediction as a deterministic process using either convolutional neural network [11] or recurrent neural network [30, 31], it could produce more damped arm movements and sometimes require an adversarial training scheme to improve the resulting gestures. Recent methods investigated this issue by developing a probabilistic framework to handle such uncertainty [2, 21]. This line of methods works by first learning a latent space for gestures and then sampling new gestures from the latent spaces given speech conditions to handle the randomness in gesture generation.

In this work, we proposed a two-stage model to address such uncertainty issues in gesture synthesis. The main purpose of the first stage is to obtain suitable feature representations of gesture motions. Here we train the VQ-VAE from gestures motions to learn a discrete codebook as our gesture representations. This is inspired by earlier retrieval-based gesture generation methods that utilize pre-defined gesture units to create gesture performance [15]. A VQ-VAE model naturally learns the prevalent gesture units by implicitly clustering input gesture data into codebook vectors. In the second stage, we learn an autoregressive model to predict the probability distribution for the next gesture token given previous tokens and speech conditions. We choose to model the autoregressive mapping with a transformer architecture [28], which is good at understanding the relationship between long-ranged elements.

In the experiment results we found that utilizing VQ-VAE help retain motion quality from the original data as well as increase the fluency of generated gestures. Moreover, since the gesture generation process is reduced to sample the next possible token in gesture codebook, we could generate multiple gesture sequences given the exact same speech. This mitigates the issue when multiple viable gestures exist for the same speech and avoids over-smooth or damped motions from deterministic mapping when using direct regression models [14].

In summary, our contribution is a novel two-stage method for co-speech gesture generation from multi-modal context information including audio and text. Firstly, we proposed to utilize a VQ-VAE model for modeling smaller gesture units as codebook vectors. Secondly, we proposed an autoregressive model based on the GPT-2 transformer to model the distribution on the discrete latent space of VQ-VAE and to sample new gesture sequences based on the

given speech context. The user evaluation results showed the proposed method can produce gesture motions with reasonable human likeness and gesture appropriateness.

## 2 BACKGROUND

### 2.1 Co-Speech Gesture Generation

There are mainly two approaches to the automatic speech gesture generation task. The first type is retrieval-based generation methods, which usually rely on a predefined set of gesture units that are created manually and the synthesis is done via keywords matching or semantic and prosody analyses to find associated gestures in the database [3, 16, 22]. Recent methods created such gesture unit database automatically from the training speech-gesture pairs [10, 13]. The synthesis stage then includes gesture attributes estimations from speech and k-nearest neighbor search to find the gesture unit that best matches the new speech content. Our design choice of utilizing the VQ-VAE model is inspired by these methods to implicitly build the gesture unit database via codebook learning.

The second type is learning-based methods which take speech-gesture pair data and learn an end-to-end model to predict co-speech gestures. Among them, some methods formulate the problem as a direct regression from speech to gestures [17]. Such methods do not explicitly handle the issue of one-to-many mapping from speech to gestures, and may require an additional discriminator to improve the synthesis results [4, 12, 30]. Recent methods model the gesture synthesis process in a probabilistic framework and can produce multiple gesture sequences from the same speech input via latent space sampling [1, 2, 5, 21, 23]. Our method follows the similar generative model architecture of latent space learning and conditional sampling for synthesizing new gestures. The main difference is that instead of modeling a continuous latent space, we utilize the *discrete* codebook learned via VQ-VAE to learn a probability distribution on latent codes by the autoregressive model. Thus the inference process is reduced to selecting the most likely latent code from the codebook based on predicted probability distribution, which naturally models the one-to-many mapping from speech to gesture in the training data.

### 2.2 Discrete Latent Space Learning

For our work, we utilized the VQ-VAE model to extract gesture units from the training gesture motions. The codebook in VQ-VAE plays an important role in unit gesture selection. In the next part of this section, we briefly explain the basic idea of VQ-VAE and its extensions.

A VQ-VAE model [27] usually consists of three parts, an encoder, a decoder, and the codebook. The encoder maps input data onto a sequence of discrete latent variables, and the decoder reconstructs the data from these discrete variables. Both the encoder and decoder use a shared codebook. VQ-VAE was first introduced for image synthesizing and compression tasks like Text-to-Image Generation. For example, Cogview [7] concatenates text tokenizer output with image codebooks to predict the next image tokens using GPT-2 model.

Currently, VQ-VAE is known to be one of the state-to-the-art generative models not only used in images but also in time-series data such as audio. Jukebox [6] utilizes VQ-VAE to generate music

singing based on. It trained multi-level VQ-VAEs to compress audio in different resolutions into discrete space and then used autoregressive Transformers to learn the latent codes for music generation. VQ-VAE was also adapted to generate repetitive rhythms of music by learning from extracted music loops. Multi-Instrumentalist Net [26] was proposed to generate multi-instrumental music from videos, which trained VQ-VAE along with an autoregressive prior conditioned on the musician's body key points movements. In our method, we aim to apply VQ-VAE in gesture-generating tasks and the evaluation results show its potential for retaining motion quality and handling probabilistic gesture synthesis.

One issue for training VQ-VAE is codebook collapse. Codebook collapse happens when the model only learns to use a small subset of the codes in the codebook, leaving a majority of the codes unused. Several methods and techniques have been proposed to prevent codebook collapse. Jukebox [6] introduced the technique of re-initializing the unused codes to a random vector to prevent dead codes during each training iteration. Video GPT [29] finds normalizing MSE for the reconstruction loss also mitigates codebook collapse. Also, some hierarchical VQ-VAEs were proposed recently for better codebook utilization. VQVAE2 uses a hierarchy of VQVAE to extract bottom and top features unconditionally, and the feedforward decoder mitigates the codebook collapse to some extent [25]. RQ-VAE uses a fixed size of codebook to recursively quantize the feature map represented as a stacked map of discrete codes, which decreases the codebook size and stabilizes the codebook training [19].

### 2.3 Multi-modal Text-to-Image Synthesis

Text-to-image synthesis is a conditional image generation task that creates images to reflect the meaning of textual descriptions. Some recent works are based on the two-stage VQ-VAE and transformer architecture. DALL-E [24] utilizes autoregressive models to process the text and image tokens as a single stream of data for image generation. They found that directly modeling the priors over raw pixels tends to prioritize short-range dependencies between pixels. To generate higher quality results, they model the priors over the latent codes extracted by VQ-VAE. Similarly, the work by Esser et al [8] use CNNs to learn a context-rich vocabulary of image constituents based on VQ-GAN and utilize transformers to efficiently model the composition with conditional context information. Our model architecture is inspired by these recent successes in image synthesis and the goal is to adapt this idea for gesture synthesis.

## 3 DATA PRE-PROCESSING

The training data for the GENEA Challenge 2022 is based on a subset of the Talking with Hands (TWH) dataset [20]. For input gesture representation, we first down-sampled input motions to 20 fps and applied a sliding window of 64 frames with 10 frames step size to produce gesture samples. Each gesture sample is converted into a tensor of size $T \times J \times D$, where $T = 64$ is the sliding window size, $J$ is the number of joints, and $D$ is the size for joint rotation representation. Following the baseline processing code, we use $J = 18$ for upper-body only gestures and $J = 24$ for full-body

gestures, excluding finger joints. We also use $D = 6$ as the representation for joint rotations based on previous research [34] to prevent singularities and reduce rotation approximation errors.

Since the dataset includes dyadic conversations instead of monologue speech as in previous gesture datasets [9, 30], we found that some portions of the training data include only listening behaviors without any speech utterances or gestures. To reduce the effects of such non-speech samples, we also filter the training data by removing samples with less than two speech words. Such data account for about 30% of all samples and removing them is more efficient for training the model.

When developing our methods, we also utilized the dataset from GENEA Challenge 2020 [18], which is based on Trinity Gesture Dataset [9]. Since the dataset includes only monologue speech gestures and is one-fourth of the TWH dataset in size, it is easier to tune and validate the models. Utilizing this data during development helps reduce the training time and allows a faster turn-around for different ideas. When training the model for the submission data, we excluded the Trinity dataset and only trained with the TWH dataset.

## 4 METHOD

The proposed method is motivated by the recent works in cross-modal text-to-image synthesis [8, 24] that utilize VQ-VAE as latent space representation for image patches and generate new images via autoregressive models to predict discrete tokens for each patch. As a high-level analogy for gesture generations, this could be seen as extracting a smaller set of gesture units from the training motions and learning the conditional probability distribution for these gesture units based on speech context and previous gestures. One motivation for learning the gesture units as discrete latent vectors is because the generation process can be seen as sampling from the codebook instead of interpolation within a continuous latent space and thus we hypothesize that the resulting gestures are more likely to retain their motion quality from original data. Moreover, learning the probability distribution in the discrete codebook will naturally handle the issue when different gesture motions are associated with the same speech in the training data since during inference the model can randomly pick one of these gesture units instead of outputting their average.

### 4.1 Overall Architecture

Our model consists of 4 components, 2 encoders for text and audio feature extraction, a VQ-VAE for gesture feature extraction, and a transformer decoder for gesture generation as shown in Figure 1. The text and audio encoders are largely based on the Trimodal model [30] while the VQ-VAE and autoregressive transformer architectures are adapted from the recent work in image synthesis from natural language [8].

As stated in the previous section, the input data is a gesture clip of 64 frames. This is done in the prepossessing step, where we use a sliding window to segment the speech and gesture into sample clips instead of producing the full speech and gesture sequence in one pass. To maintain the continuity of output gestures, we include a 10-frame overlap between each clip for consecutive syntheses.

Our model includes two stages. The first stage only involves training the VQ-VAE model to learn discrete feature representation. In the second stage, we will freeze the weight of VQ-VAE to treat it as a gesture encoder and use the transformer to learn the probability distribution over the discrete latent space.

### 4.2 Learning discrete feature representation

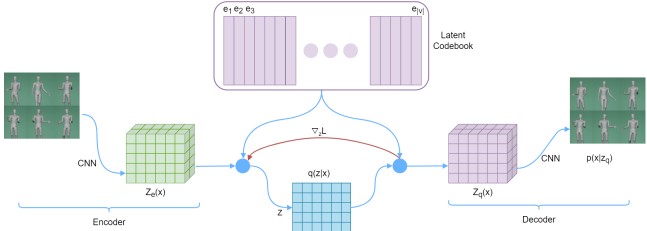

**Figure 1: VQ-VAE architecture**

The first stage of our learning process is to train the VQ-VAE model to extract small gesture units as tokens from raw gestures. Since it is an autoencoder architecture, both the input and output for VQ-VAE include only gesture samples. In the training process, each gesture sample $x$ is a tensor of size $T \times P$, where $T$ is the number of frames per clip and $P = J \times D$ is the pose feature size. The encoder first downsamples the input gesture into the tensor $Z_e(x)$ with size $t \times p$. Then each $p$-dimensional vector from $Z_e(x)$ is quantized to the nearest embedding in a learnable codebook $V = \{e_1, e_2, ..., e_{|V|}\}$, with embedding dimension $p$ and codebook size $|V|$. Specifically, during the quantization stage each feature vector from the encoder output $Z_e(x)$ is replaced by the index of the nearest vector $e_k$ in the codebook. The quantization step can be summarized as:

$$Quantize(Z_e(x)) = e_k \ where \ k = \underset{j}{argmin}||Z_e(x) - e_j||$$

During the reconstruction stage, the decoder takes the quantized latent vector $e$ and maps it back to the reconstructed gesture $\hat{x}$ in the original dimension. Besides the $L1$ reconstruction loss, VQ-VAE also includes two additional loss terms. The codebook loss helps the codebook variable training and the commitment loss updates encoder weights. The objective function for training the VQ-VAE is then defined as the following:

$$L(x, \ \hat{x}) = ||x - \hat{x}|| + ||sg[Z_e(x)] - e||_2^2 + \beta||sg[e] - Z_e(x)||_2^2$$

The operator $sg$ refers to the stop-gradient operator and $\beta$ is a hyperparameter that controls the weight of commitment loss. The quantized result can be represented by $t$ indices of the vector in the codebook. The decoder maps and upsamples the quantized vectors back to reconstruct the original input. In our model, we use $t = 16, d = 128$ in the encoder for feature extraction, and $|V| = 1024$ for codebooks size in full-body model and $|V| = 128$ for upper-body only model.

One issue for training VQ-VAE is the codebook collapse. This problem happens when only a small subset of codes are utilized in the codebook during training and will result in a latent space with less representational power. To prevent codebook collapse, we

utilized several strategies proposed in previous work for learning VQ-VAE. First, we apply exponential moving averages for the codebook learning, which places a greater weight update on the most recent codebook vectors [27]. In addition, we reset the codes that are not used to random values to allow them a better chance to be utilized in the next iterations, as proposed in the Jukebox paper [6]. We also found that dividing the variance of the dataset when calculating MSE reconstruction loss does have a small improvement for the training [29].

## 4.3 Training auto-regressive model

The input conditions for our auto-regressive model are raw audio and text. Our text and audio encoders are inspired by the Trimodal [30], which uses the 1D-convolutional encoder for audio data and temporal convolutions for text tokens of word embedding. After encoding data from each modality through separate encoders, we concatenate both outputs into a single condition feature tensor $f_c$ with size $T \times (d_1 + d_2)$, where $d_1$ and $d_2$ are the encoder output dimensions for audio and text.

In the training process, we aim to predict the next gesture token based on previous tokens and condition vectors. We take the feature vectors for text and audio as the condition on the transformer and do left-to-right prediction of tokens, similar to other language modeling tasks on the discrete latent codes. To simplify the architecture, we treat both condition vectors and gesture tokens the same way as transformer inputs. The transformer we use is similar to conditioned GPT-2 with masked self-attention. For each training sample, the transformer predicts a sequence of gesture token at every position after the condition vectors.

Specifically, during training the model takes the condition features $f_c$ and previous gesture tokens $g_{1:t}$ as input to predict the probability distribution of the next token $p_i(t+1)$ for each code $e_i$ in the codebook. Here the autoregressive steps at each position $t = 1, \ldots, \frac{T}{s}$ sequentially predict the gesture tokens within the sliding window, where $s$ is the downsampling factor in the VQ-VAE encoder and $T = 64$ is the sliding window size. In our experiment, we set $s = 4$ to allow each gesture token to represent about 0.2 second of gesture motion. The loss is calculated using negative log-likelihood at each position.

When inferring from longer speech, we predict T frames at a time and merge them into the final gesture sequence. Each sliding window has a 10 frames overlap with the previous window and the poses are interpolated in the overlapped area to maintain motion continuity. More specifically, for simplicity, we discard last 10 frames of the follow-up predicted sequence and merge them together, and there is no affine combination or decaying of interpolation was applied. The gesture tokens within each sliding window are predicted in a similar autoregressive manner by inferring the probability $p_i(t+1)$ for the next token. To allow more variety in the resulting gestures, instead of selecting the latent code with the highest probability from the codebook, we randomly sample it using the top-k probabilities where $k = 10$ for both upper-body and full-body gesture generations.

## 5 EVALUATION

The main evaluations were performed by the organizers and included both subjective human studies and objective metrics [32]. Since the overall quality of the generated gestures is more subjective and includes multiple aspects that are difficult to quantitatively measure such as fluency, consistency, or whether the generated gestures match with the speech, the evaluation process placed more emphasis on human studies. For the subjective evaluation, two different human study tasks were performed to compare the pros and cons of different methods. The Human-likeness study measures how much the generated gestures look like the motion of a real human while ignoring the effect of the speech. On the other hand, the appropriateness study measures whether the generated gestures match the speech content. To avoid bias toward more realistic motions, the generated gesture is compared against a random motion from the same submission.

For each team, 40 chunks from their submitted motions were selected and evaluated for both full-body and upper-body only gestures. Figure 3 and 4 provide visualizations to summarize the user study results among different submissions and baselines. Our submission is FSI entry in the full-body study and USJ entry in the upper-body study. As shown in the study results, our method was able to achieve relatively good performance in both human-likeness and appropriateness tasks.

## 6 RESULTS AND DISCUSSION

From the user evaluation results, our method performs relatively well for human likeness and is among top-3 methods in all submissions for both upper-body only and full-body results. Since we trained two separate VQ-VAE and autoregressive models from upper-body and full-body data, this consistent result validates our goal of utilizing VQ-VAE to extract gesture units that retain motion quality from the original data. However, we also notice that there is still a significant quality gap between our synthesis results and ground truth gesture motions. There are two possible explanations for the gap. Firstly, since the method processes one sliding window at a time and interpolates adjacent windows to form the gesture sequence, there could be discontinuities between two adjacent windows. While interpolating data in the overlapped frames removes the pose discontinuity, it does not resolve the velocity and acceleration discontinuity. Such higher order discontinuities produce more jerky movements periodically and will likely reduce the overall gesture quality. Secondly, the autoregressive model may not predict the correct gesture tokens and some bad tokens might disrupt the quality of decoded gesture motions as seen in the results of previous text-to-image synthesis works.

Another issue we noticed when visualizing our synthesis results is that for the full-body gestures, our method tends to produce a lot of weight-shifting movements in the lower body. While the movement itself is not unnatural, the frequent lower body movements may look distracting and be regarded as motion artifacts in the user evaluations. This might also explain the slightly lower score for our full-body evaluations. We believe this issue might due to the fact that we encode all joints in the full-body poses into a single gesture token, and thus without proper constraints, the predicted gesture tokens may include any lower body movements as far as they are

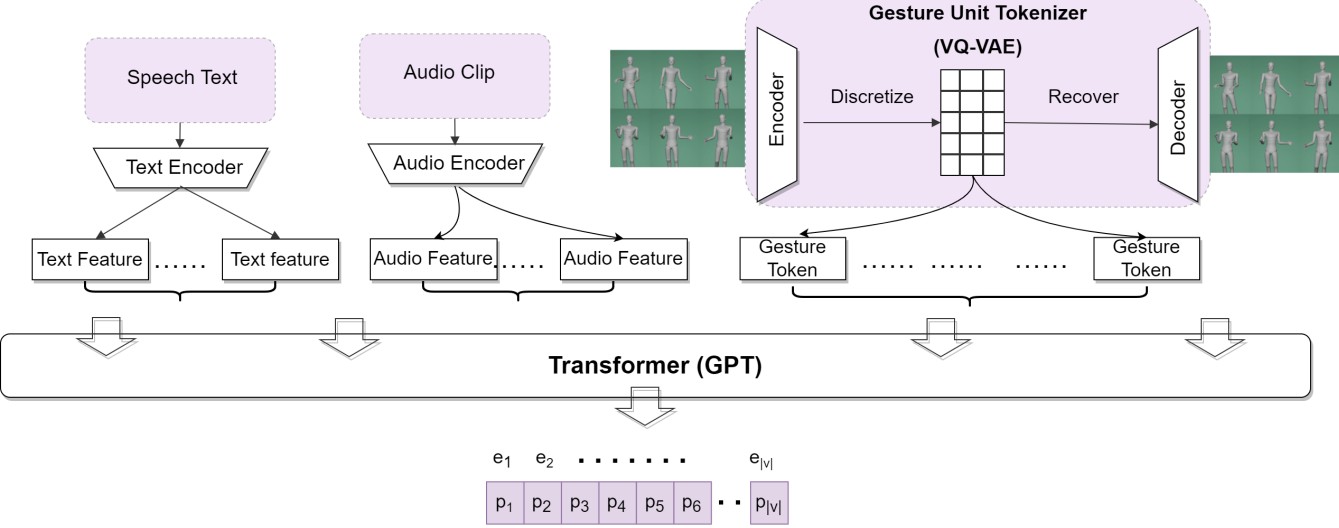

Figure 2: The autoregressive transformer for predicting the probablity for the next gesture token.

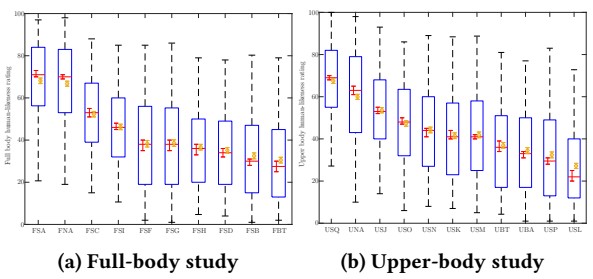

**(a) Full-body study**    **(b) Upper-body study**

Figure 3: Box plot visualization for the human-likeness studies.

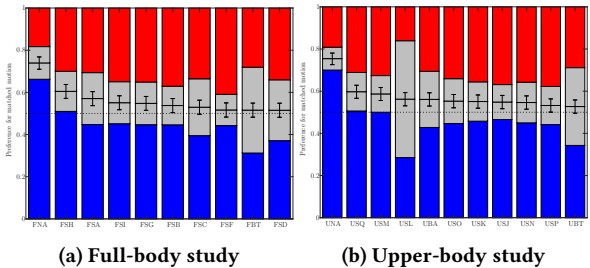

**(a) Full-body study**    **(b) Upper-body study**

Figure 4: Bar plot visualization for the gesture appropriateness studies.

deemed likely based on input speech conditions. Separating both the VQ-VAE and autoregressive models into a multi-level skeleton hierarchy may alleviate this problem as we could learn different probability distributions for different parts of the body.

The evaluation for appropriateness shows that the differences are relatively small between all submissions and the ground truth

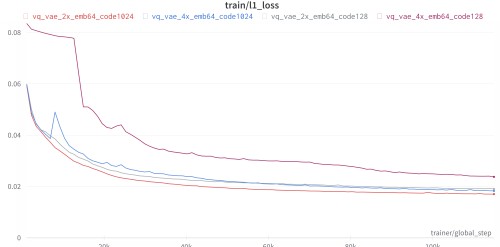

Figure 5: Plot of reconstruction loss when training VQ-VAE with different codebook sizes and downsample factors. With small codebook size ($|V| = 128$ and $s = 4$), reconstruction error is significantly higher than model trained larger codebook size ($|V| = 1024$).

result is leading by a large margin. Although our method performs consistently fine in the human-likeness evaluations, our results for appropriateness are mixed. In the full-body study, our result is still in the top 3 among all submissions, but for upper-body evaluations, our method produces one of the lowest results. While the gap is not significant, we find this result puzzling initially as the upper-body movements should be easier to model by the autoregressive model. After further investigations, we found that this might be due to the codebook size $|V|$ we chose for the upper-body only model. As shown in Figure 5, our setting of $|V| = 128$ for the upper-body only model has a gap in reconstruction loss when compared with the model trained with a larger codebook size $|V| = 1024$. This might indicate that our upper-body setting is not able to fully capture the complete gesture space of the training data. As a result, the autoregressive model is not learning all the richness of gesture motions and might produce simpler gestures with more

limited gesture tokens. Our findings after further reviewing the visualization results support this guess as our upper-body only model tends to repeat similar gestures multiple times for the whole utterances and the more interesting co-articulated gestures are less frequent when compared with the full-body model. Thus while our upper-body model produces results with reasonable human-likeness from gesture tokens, it is not able to produce gestures with enough diversity to account for different speech utterances and thus might result in a lower appropriateness score. This finding also shows that for our method, training a good VQ-VAE model is vital as it could impact both the reconstructed motion quality and the richness of the resulting autoregressive models.

## 7  CONCLUSIONS AND TAKEAWAYS

In this work, we proposed a two-stage approach that utilizes VQ-VAE for learning gesture units and an autoregressive transformer for learning conditional latent code priors. The evaluation results show the potential of the method for generating gestures with adequate human-likeness. While our method performed relatively well in the user evaluation studies, we did not have enough time to implement and experiment with different variations of our method to produce more refined results. Specifically, there are a few issues we would hope to investigate and further improve the system.

One key issue is that the VQ-VAE training is not as stable as we originally expected due to codebook collapse. Thus we would hope to develop a more robust process for training the VQ-VAE, especially for different datasets. During our initial development, we used the Trinity dataset [9] to allow faster iterations in model training and parameter tuning. However, when we switched to the TWH dataset and applied the same hyper-parameters ($|V| = 64$ and $s = 4$) that were working well on the Trinity dataset, we found that the results were worse than expected and we had to re-do the parameter tuning to build an acceptable latent cookbook. Since the codebook collapse issue is still not fully solved for VQ-VAE, this might explain why the results are not as stable across different datasets with a varying number of joints and motion quality. We also hope to investigate newer techniques [33] that mitigate the codebook collapse issue to make the training less vulnerable to hyper-parameter changes.

We would also hope to improve the autoregressive model to address the sequential nature of gesture motions. Due to the timeline for the challenge, we simplified the transformer implementation and adapted the typical architecture from image synthesis for our method. Thus it only works at a fixed-size sliding window without considering longer gesture sequences. This not only makes the system less flexible but also introduces potential artifacts between adjacent windows. A specialized model for handling time-series data of arbitrary length should work better for gesture synthesis.

Finally, in our submission, we were not able to model the root translations and for simplicity, we fixed the root joint positions for all of our results. Thus the lower body movements from our submission are less natural compared with the other top results. Implementing the techniques of full-body synthesis such as the one proposed in [2] to handle the root translations will enhance our method to generate long sequences of full-body gestures with realistic lower body motions.

## ACKNOWLEDGMENTS

The projects or efforts depicted were or are sponsored by the U. S. Army. The content or information presented does not necessarily reflect the position or the policy of the Government, and no official endorsement should be inferred.

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
