# OpenReview forum: "The DeepMotion entry to the GENEA Challenge 2022"
_ACM.org/ICMI/2022/Workshop/GENEA — GENEA Challenge & Workshop 2022 Mainproceeding_

### Official Review · Reviewer_NP7P · 2022-07-31
**A sound and well-motivated approach that performs well on the evaluation benchmarks**

**Rating:** 7
**Confidence:** 5

**Review:**

## Strengths
- The proposed methodology of using VQ-VAE-based gesture tokens to train a GPT2-Transformer-based mapping from audio to gestures is sound and well-motivated.
- The paper is well-written and easy to follow.
- The approach shows good performance on the evaluation benchmarks.

## Weaknesses
- Missing references:

[A] Taras Kucherenko, Patrik Jonell, Sanne van Waveren, Gustav Eje Henter, Simon Alexandersson, Iolanda Leite, and Hedvig Kjellström. "Gesticulator: A framework for semantically-aware speech-driven gesture generation." In Proceedings of the 2020 International Conference on Multimodal Interaction, pp. 242-250. 2020.

[B] Uttaran Bhattacharya, Nicholas Rewkowski, Abhishek Banerjee, Pooja Guhan, Aniket Bera, and Dinesh Manocha. "Text2gestures: A transformer-based network for generating emotive body gestures for virtual agents." In 2021 IEEE Virtual Reality and 3D User Interfaces (VR), pp. 1-10. IEEE, 2021.

[C] Uttaran Bhattacharya, Elizabeth Childs, Nicholas Rewkowski, and Dinesh Manocha. "Speech2affectivegestures: Synthesizing co-speech gestures with generative adversarial affective expression learning." In Proceedings of the 29th ACM International Conference on Multimedia, pp. 2027-2036. 2021.

- It is not clear to me how the interpolation works on the overlapping frames between sliding windows (Ln. 308-311):
    - What is the interpolation methodology? For example, if it is an affine combination, what are the per-term coefficients?
    - Does the interpolation only take place on the overlapping segment? Or does it also propagate (e.g., with some decaying of the interpolation coefficients) to the subsequent frames? In other words, how do the authors ensure there are no discontinuities at the interpolation boundaries?

## Minor comments
- The paper has some grammatical inconsistencies, such as
    - Ln. 257: "we applies" -> "we apply"
    - Ln. 259: "a random values" -> "random values"
    - Ln. 293: "based previous tokens" -> "based on previous tokens"
    - Ln. 308: "When inference" -> "When inferring"

I recommend another round of proofreading the paper.

## Overall
The paper presents a detailed, sound, and well-motivated approach to synthesizing co-speech gestures. The results are promising and have performed well on the evaluation benchmarks. For completeness, please add the missing references and clarify the interpolation procedure.

---

### Official Review · Reviewer_EMmU · 2022-08-08
**The DeepMotion entry to the GENEA Challenge 2022**

**Rating:** 9
**Confidence:** 4

**Review:**

The paper introduces a framework inspired by the VQ-VAE model and the GPT-2 transformer to generate body gestures. In the first stage, the VQ-VAE network is trained to extract gesture tokens from raw input gestures. In the second stage, gesture tokens, text, and audio features are fed to GPT-2 model to produce body motion outputs.

Strength:
1. This is a well-investigated paper. The proposed approach is also interesting and novel. The authors have implemented the VQ-VAE model into the gesture generation domain. The GPT-2 transformer is also an interesting implementation.

2. The paper is very well-written and organized. The explanations of the designed framework are sufficient. It allows readers to fully understand the techniques implemented and possibly, reproduce the results.

Weakness:
I would suggest several points that the authors may consider in the final paper version as well as future works:

1. Visualization results - There are several discussions in Section 6. Results and Discussion, for example, "our method tends to produce a lot of weight shifting movements in the lower-body. (L393-394)". In this case, it would be more informative if the authors could include an example of generated gestures to support that statement.

2. Further analysis - an ablation study could be included to better verify the model components, for example, the codebook size. Indeed, several objective metrics (introduced in the previous GENEA workshops) can be implemented rather than relying on the reconstruction loss for tuning the codebook size. I assume that the objective metrics would allow the authors to have better ideas about the quality of generated motion.

3. Technical concerns - 3D CNN for encoding/decoding motions: although CNN can also be implemented for the time series data, the placement of joint and time sequence in CNN should be carefully designed to ensure the spatial-temporal presentation of motion.

**Nominate For A Reproducibility Award:**

The explanations of the designed framework are sufficient for readers to understand the techniques implemented and possibly reproduce the results. I would like to recommend this work for a reproducibility award if the authors can submit their code and related data in the final submission.

---

### Official Review · Reviewer_Tzsn · 2022-08-09
**This paper presents a new method for co-speech gesture generation using VQ-VAE to learn a probability distribution on latent codes, which naturally models the one to many mapping from speech to gesture in the training data.**

**Rating:** 7
**Confidence:** 5

**Review:**

Well written paper with some good ideas. Nice to see a VQ-VAE approach being used for this problem. While the results are still not reaching close to ground truth gesture motions, it is a promising approach and the authors have a good discussion on the limitations and potential solutions, given additional time.

It will be interesting in future work to see if the authors can overcome the issues of codebook collapse using more robust training, etc.

---

### Decision · Program_Chairs · 2022-08-11

**Decision:**

Accept (Main proceeding)

**Comment:**

All the reviewers recommended accepting this paper. The reviewers agreed that the proposed approach is novel and the paper is well written overall. Based on the reviews, the chairs accept this paper for publication in the main ACM ICMI proceedings.

Please read the reviews carefully and revise the paper for the camera-ready version as follows:

1. Proof-read and fix grammatical errors.
2. Include the missing details of the motion interpolation between consecutive motion blocks.
3. Consider including references suggested by the reviewer NP7P.
4. Consider sharing videos demonstrating the results (weakness as well)
5. Weaknesses 2 and 3 as commented on by reviewer EMmU are also relevant, and the paper will be improved if you can address these in the revision. You don’t have much time to revise the paper, so we understand if that doesn’t happen.